

# The effect of the COVID-19 pandemic on health care workers' anxiety levels: a meta-analysis

Lunbo Zhang[1], Ming Yan[1], Kaito Takashima[1], Wenru Guo[1] and Yuki Yamada[2]

[1] Graduate School of Human-Environment Studies, Kyushu University, Japan
[2] Faculty of Arts and Science, Kyushu University, Japan

Corresponding author
Lunbo Zhang,
zhang.lunbo.186@s.kyushu-u.ac.jp

## ABSTRACT

**Background:** The COVID-19 pandemic has been declared a public health emergency of international concern, causing excessive anxiety among health care workers. Additionally, publication bias and low-quality publications have become widespread, which can result in the dissemination of unreliable information. A meta-analysis was performed for this study with the following two aims: (1) to examine the prevalence of anxiety among health care workers and determine whether it has increased owing to the COVID-19 pandemic and (2) to investigate whether there has been an increase in publication bias.

**Methods:** All relevant studies published between 2015 and 2020 were searched in electronic databases (namely Web of Science, PubMed, Embase, PsycInfo, PsyArXiv, and medRxiv). The heterogeneity of the studies was assessed using the $I^2$ statistic. The effect size (prevalence rate of anxiety) and 95% CI for each study were also calculated. We used moderator analysis to test for the effect of the COVID-19 pandemic on health care workers' anxiety levels and to detect publication bias in COVID-19 studies. We assessed publication bias using funnel plots and Egger's regression.

**Results:** A total of 122 studies with 118,025 participants met the inclusion criteria. Eighty-eight articles (75,066 participants) were related to COVID-19, 13 articles (9,222 participants) were unrelated to COVID-19 (*i.e.*, articles related to other outbreaks, which were excluded), and 21 preprints (33,737 participants) were related to COVID-19. The pooled meta-analysis prevalence was 33.6% (95% CI [30.5–36.8]; 95% PI [6.5–76.3]). Moderator analysis revealed no significant differences between articles related to COVID-19 and those unrelated to COVID-19 ($p = 0.824$). Moreover, no significant differences were found between articles and preprints related to COVID-19 ($p = 0.843$). Significant heterogeneity was observed in each subgroup. An Egger's test revealed publication bias in both articles and preprints related to COVID-19 ($p < 0.001$).

**Conclusions:** Determining whether the anxiety state of health care workers is altered by the COVID-19 pandemic is currently difficult. However, there is evidence that their anxiety levels may always be high, which suggests that more attention should be paid to their mental health. Furthermore, we found a substantial publication bias; however, the quality of the studies was relatively stable and reliable.

# INTRODUCTION

## Background and research questions

The COVID-19 pandemic has affected more than 18.9 million individuals and resulted in over 709,000 deaths globally (*World Health Organization, 2020a*). Therefore, it has been declared a public health emergency of international concern (*World Health Organization, 2020b*). To overcome this crisis, it is important to maintain an adequate health care workforce, which requires not only a sufficient number of health care workers but also the maximization of each health care worker's ability to care for a greater number of patients. As outbreaks could last several years, it is critical that health care workers are able to perform to their full potential over extended periods (*Shanafelt, Ripp & Trockel, 2020*).

The COVID-19 pandemic has affected many aspects of people's lives, particularly their mental health (*Wang et al., 2020*; *Özdin & Bayrak Özdin, 2020*; *Mazza et al., 2020*; *González-Sanguino et al., 2020*). While health care workers have to cope with the societal shifts and emotional stressors faced by the general population concurrently, they also face greater risks of exposure, extreme workloads, moral dilemmas, and rapidly evolving practice environments that differ greatly from what they are familiar with (*Adams & Walls, 2020*; *Xiang et al., 2020*). Moreover, the previously unknown challenges in terms of both physical and mental health causes excessive tension and anxiety among health care workers (*Albott et al., 2020*). While anxiety is a common mental condition that can cause emotional distress, obsessive thinking, and compulsive behavior, long-term anxiety results in psychological distress and even affects individuals' daily lives (*Leeming, Madden & Marlan, 2014*). Anxiety also impairs the executive functions that underlie one's ability to control and focus on their thoughts (*Shields et al., 2016*). Consequently, studying and accurately grasping health care workers' anxiety levels is necessary to take more appropriate and corrective measures to manage public health and safety.

Although some researchers have investigated health care workers' anxiety levels during the COVID-19 pandemic (*Pappa et al., 2020*; *Pan, Zhang & Pan, 2020*), many new papers on COVID-19 have been released rapidly as the pandemic continues to pose a serious threat. The present meta-analysis included the latest papers and aimed to generate a more comprehensive understanding of the prevalence of anxiety among health care workers. Furthermore, to date, no comparison has been conducted between studies on health care workers' anxiety levels related to the COVID-19 pandemic and those unrelated to the pandemic. In the context of the current outbreak, would studies conducted in two different periods (*i.e.*, during COVID-19 and during normal times) have different effect sizes? Would anxiety levels increase significantly? Accordingly, the first aim of our meta-analysis was to examine health care workers' anxiety status and determine the influence of the COVID-19 pandemic by comparing COVID-19-related studies to unrelated studies. Our purpose was to explore the impact of the COVID-19 pandemic on the anxiety level of health care workers, that is, whether their anxiety levels have increased from previous

levels. To avoid contamination from other outbreaks on health care workers' anxiety during time periods prior to the COVID-19 pandemic, in this study, we excluded articles related to other outbreaks from our dataset of studies unrelated to COVID-19.

In addition, since the onset of the outbreak, there has been a dire need for knowledge regarding COVID-19, and medical journals have drastically accelerated the publication process for COVID-19-related articles to facilitate knowledge acquisition (*Palayew et al., 2020*; *Horbach, 2020*). In this situation, the preference for publishing studies with significant results is more extreme, which may seriously compromise the ability to draw valid conclusions from published literature. As publication bias may be a significant flaw, the second aim of our meta-analysis was to investigate it by comparing unpublished preprints on COVID-19 to published journal articles on this topic.

We believe that this study can contribute to better supporting the mental health of health care professionals by identifying their anxiety levels during the COVID-19 pandemic. In addition, by examining the publication bias of articles published during COVID-19, we can raise the issue of their quality with the psychological and medical communities and contribute to reducing publication bias in the future.

### Hypotheses

For this study, we generated the following hypotheses:

1. The COVID-19 pandemic makes health care workers more anxious; thus, studies related to COVID-19 have a larger effect size. We investigated this by comparing studies related to COVID-19 to unrelated studies.
2. Publication bias in COVID-19-related studies is widespread. We investigated this by comparing unpublished preprints on COVID-19 to published journal articles on the disease.

## MATERIALS AND METHODS

### Preregistration

The research protocol for this study was peer-reviewed and registered prior to data collection at https://www.researchprotocols.org/2020/11/e24136.

### Search strategy

This study followed the PRISMA (Preferred Reporting Items for Systematic Reviews and Meta-Analyses) guidelines (*Page et al., 2021*). We searched electronic databases—Web of Science, PubMed, Embase, PsycInfo, PsyArXiv, and medRxiv—up to November 9, 2020, for all published journal articles (related *vs.* unrelated to COVID-19) and preprints (relevant to COVID-19), the titles and abstracts of which included the search terms presented in Textbox 1.

### Inclusion and exclusion criteria

Studies were included only if they met the following inclusion criteria: (1) written in English (which was decided based on the research team's unified considerations);

> **Textbox 1 Search terms.**
>
> ("Health Personnel" OR "Personnel, Health" OR "Health Care Providers" OR "Health Care Provider" OR "Provider, Health Care" OR "Providers, Health Care" OR "health care Providers" OR "health care Provider" OR "Provider, health care" OR "Providers, health care" OR "health care Workers" OR "health care Worker" OR "Health Occupations" OR "Health Occupation" OR "Health Professions" OR "Health Profession" OR "Profession, Health" OR "Professions, Health" OR "Health professions")
>
> AND
>
> (Anxiety OR Hypervigilance OR Nervousness OR "Social Anxiety" OR "Anxieties, Social" OR "Anxiety, Social" OR "Social Anxieties")

(2) related to "anxiety among health care workers"; (3) utilized quantitative research designs; (4) were submitted during the period of 2015–2020; (5) included standardized measures of anxiety with published psychometric data and reasonable evidence of reliability and validity; (6) included a clear description of the methods used to assess and score standardized measurement instruments; and (7) included publicly available effect sizes (prevalence) or values that could be calculated (the number of health care workers with anxiety and the sample size).

The exclusion criteria were as follows: (1) studies with insufficient data; (2) duplicate sources; (3) studies with unclear methods; and (4) publications on other outbreaks.

## Data extraction

First, duplicate studies found in multiple databases were excluded. Subsequently, titles and abstracts were screened and papers were removed based on the inclusion and exclusion criteria. The full text of the articles was then checked, and article information was extracted using a pre-prepared extraction table that included the title, authors' names, scales used, diagnostic cut-off, year of submission, country, and sample size as well as whether the study has been published, whether the study was related to COVID-19, and the study's effect size (prevalence of anxiety), WHO regions, World Bank income groups, study design, data collection, participants' mean age, proportion of females, marital status, work status, type of hospital, professional role, preparedness of countries in terms of hospital beds per 10,000 people, stringency index during the survey, and human development index. Two of the authors (L.Z. and M.Y.) independently performed the article review and data extraction. When there was disagreement between them, the remaining authors resolved the conflict.

## Study assessment criteria

We used the Strengthening the Reporting of Observational Studies in Epidemiology (STROBE) checklist to assess the quality of the observational studies (*Sanderson, Tatt & Higgins, 2007*). The checklist consists of six scales: title, abstract, introduction, method, results, and discussion, each of which includes multiple items, comprising a total of 32. Each item is scored as 0 (not fulfilled) or 1 (fulfilled). In the modified STROBE, scores ranged from 0 to 32, with scores ≥16 indicating a low risk of bias and scores <16 indicating a high risk of bias. Studies that exhibited a low risk of bias were selected for analysis.

## Statistical analysis

We used Stata version 17 for all analyses and generated forest plots of the summary pooled prevalence. First, we used the Freeman–Tukey double arcsine method to transform our data before the heterogeneity of the studies was determined using the $I^2$ statistical index, which ranged from 0 to 100, with higher scores indicating greater heterogeneity. The categories encompassed by the index were defined based on the test developed by *Higgins et al. (2003)* to measure the extent of heterogeneity: low (25%), moderate (50%), and high (75%). A study with heterogeneity >50% prompted the use of random-effects models. For each study, we calculated the effect size (prevalence rates of anxiety), 95% confidence interval (CI) around the effect size, and 95% prediction interval (PI). If the original paper did not include the effect size or the number of health care workers with anxiety (which can be used to calculate the effect size), the authors of the paper were contacted and asked to provide this information. If they were unable to do so, the study was excluded from analysis.

Subsequently, we used moderator analysis (meta-regression) to test for the effect of COVID-19 on health care workers' anxiety levels (related *vs.* unrelated to COVID-19) and publication bias in COVID-19 studies (preprints *vs.* published journal articles). To further explore the effect of participants and study characteristics on the prevalence estimates, we performed additional subgroup analyses and meta-regression on the factors that had at least five studies at each level of the ranges. Publication bias was assessed using funnel plots and Egger regression (*Egger et al., 1997*). For Egger regression, a *P* value less than the significance level (α = 0.05) suggested that publication bias was present. If publication bias was present and studies had no homogeneity, the trim-and-fill procedure was applied to adjust for these missing studies (*Duval & Tweedie, 2000*).

Finally, sensitivity analyses (leave-one-out) were performed to assess the influence of each study on the pooled effect size. The statistical significance level was set at α = 0.05 (*Bouras et al., 2019*).

# RESULTS

## Study selection and characteristics

The initial search resulted in 4,174 articles and 2,223 preprints. After removing duplicates and excluding ineligible studies, 101 articles (88 related to COVID-19 and 13 unrelated to COVID-19) and 21 preprints (related to COVID-19) were ultimately included (*AlAteeq et al., 2020; Almater et al., 2020; Alshekaili et al., 2020; Apisarnthanarak et al., 2020; Arafa et al., 2021; Azoulay et al., 2020; Badahdah et al., 2020; Ayhan Başer et al., 2020; Cai et al., 2020; Cai et al., 2020; Chen et al., 2020; Chen et al., 2021; Chew et al., 2020; Chew et al., 2020; Dal'Bosco et al., 2020; Di Tella et al., 2020; Dobson et al., 2021; Elbay et al., 2020; Elhadi et al., 2020; Elkholy et al., 2020; Erquicia et al., 2020; Firew et al., 2020; Giardino et al., 2020; Gupta et al., 2020; Hasan et al., 2020; Holton et al., 2020; Huang & Zhao, 2020; Huang et al., 2020; Johnson, Ebrahimi & Hoffart, 2020; Kannampallil et al., 2020; Nguépy Keubo et al., 2021; Khanal et al., 2020; Koksal et al., 2020; Korkmaz et al., 2020; Kurt, Deveci & Oguzoncul, 2020; Lai et al., 2020; Li et al., 2020; Liang et al., 2020; Lu et al.,*

*2020; Luceño-Moreno et al., 2020; Magnavita, Tripepi & Di Prinzio, 2020; Mahendran, Patel & Sproat, 2020; Malgor et al., 2020; Metin, Turan & Utlu, 2020; Monterrosa-Castro, Redondo-Mendoza & Mercado-Lara, 2020; Ng et al., 2020; Ng et al., 2020; Ning et al., 2020; Pan et al., 2020; Park et al., 2020; Pouralizadeh et al., 2020; Prasad et al., 2020; Que et al., 2020; Sagaon-Teyssier et al., 2020; Şahin et al., 2020; Salopek-Žiha et al., 2020; Saricam, 2020; Shechter et al., 2020; Si et al., 2020; Temsah et al., 2020; Teo et al., 2020; Uyaroğlu et al., 2020; Veeraraghavan & Srinivasan, 2020; Wang et al., 2020; Wang et al., 2020; Wang et al., 2020; Wang et al., 2020; Wańkowicz, Szylińska & Rotter, 2020; Woon & Tiong, 2020; Xiao et al., 2020; Xing et al., 2020; Xiaoming et al., 2020; Yáñez et al., 2020; Yang et al., 2021; Zhang et al., 2020; Zhang et al., 2020; Zhou et al., 2020; Zhu et al., 2020; Li et al., 2016; Liu et al., 2017; Paiva, Martins & Paiva, 2018; Picakciefe et al., 2015; Shi et al., 2020; Tangiisuran et al., 2018; Weaver et al., 2018; Winning et al., 2018; Abid et al., 2020; Ahn et al., 2020; Alonso et al., 2020; Bachilo et al., 2020; Chung et al., 2020; Drager et al., 2020; Gilleen et al., 2020; Greene et al., 2020; Hassannia et al., 2020; Kaveh et al., 2020; Lee et al., 2020; Liu et al., 2020; Naser et al., 2020; Rossi et al., 2020; Salman et al., 2020; Tasnim et al., 2020; Thapa et al., 2020; Thomaier et al., 2020; Wanigasooriya et al., 2020; Weilenmann et al., 2020; Xu et al., 2020*). A flow diagram of the search process is presented in Fig. 1. The characteristics of the included studies are summarized in the raw data.

## Prevalence of anxiety among health care workers

The reported prevalence of anxiety among health care workers in individual studies ranged from 3.4% to 87.3%, with a pooled meta-analysis prevalence of 33.6% (95% CI [30.5–36.8]; 95% PI [6.5–76.3]) and significant between-studies heterogeneity ($p < 0.001$, $I^2 = 99.2\%$; Fig. 2).

## Moderator analysis among subgroups

We considered three subgroups: articles related to COVID-19, those unrelated to COVID-19, and preprints. The pooled prevalence of anxiety in different subgroups is shown in Fig. 2. For articles related to COVID-19, the pooled prevalence was 33.6% (95% CI [29.7–37.6]; 95% PI [6.5–73.1]), while it was 32.3% (95% CI [25.6–39.5]; 95% PI [13.6–83.8]) in articles unrelated to COVID-19 and 34.5% (95% CI [27.9–41.6]; 95% PI [3.8–84.2]) in preprints related to COVID-19. We used meta-regression, the most general and flexible method of moderator analysis (*Holden et al., 2017*), to test the effect of COVID-19 on health care workers' anxiety (related *vs.* unrelated to COVID-19) and to identify publication bias in COVID-19 studies (preprints *vs.* published journal articles). We defined articles related to COVID-19 as the reference group and conducted a meta-regression. No significant difference was found between the articles related to COVID-19 and those that were unrelated ($p = 0.824$). Moreover, no significant difference was found between the articles related to COVID-19 and the preprints ($p = 0.843$).
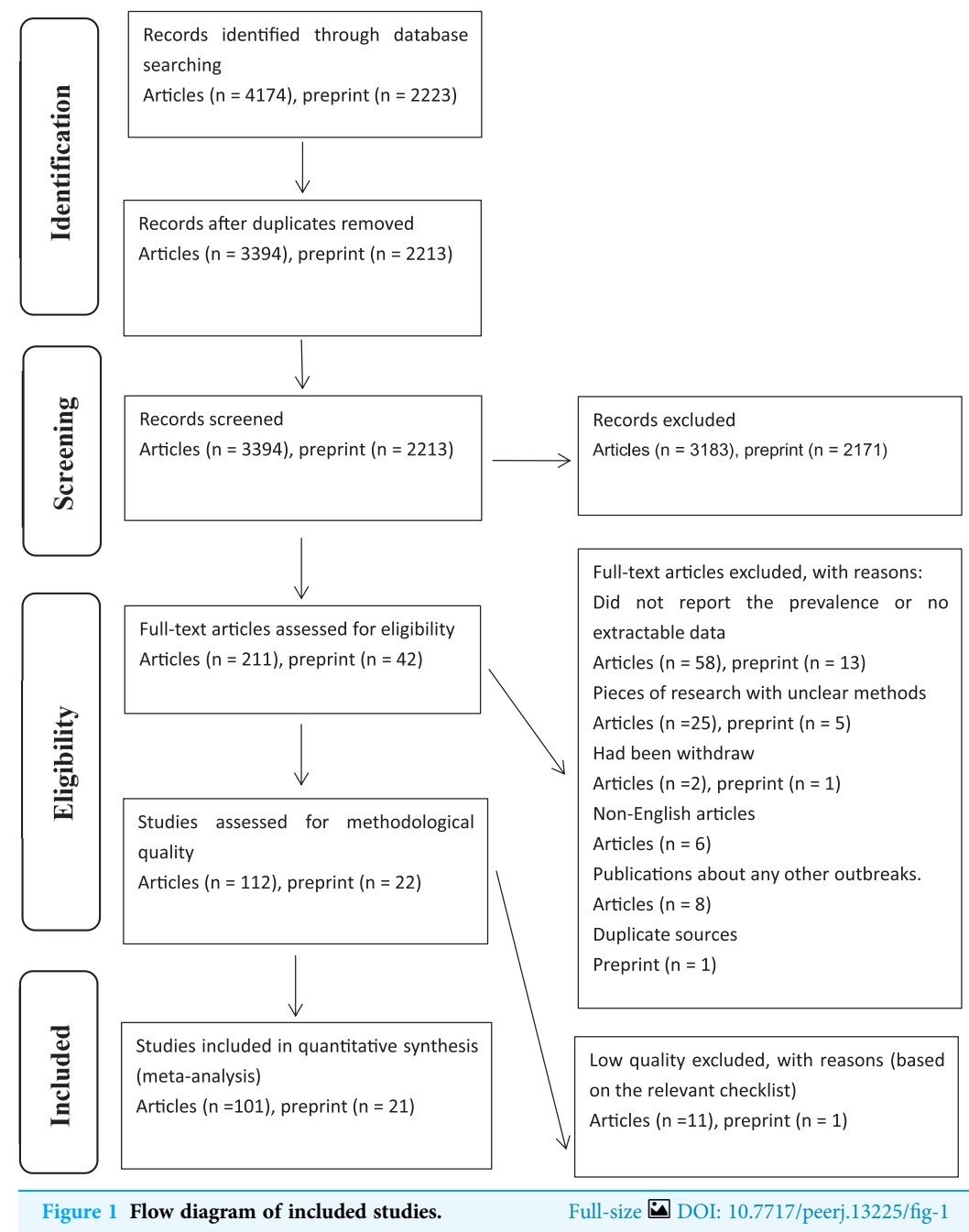

Figure 1  Flow diagram of included studies.               

## Explore the effect of participant and study characteristics

We performed subgroup analyses and meta-regression to explore the effects of participant and study characteristics on the prevalence estimates. When the meta-regression was limited to five or more studies at each level, only four factors (the proportion of females, the COVID-19 government response stringency index during the survey, the preparedness of countries in terms of hospital beds per 10,000 people, and the Human Development Index) were analyzed. Table 1 presents the results. Unfortunately, we failed to identify any factors affecting the prevalence estimates.

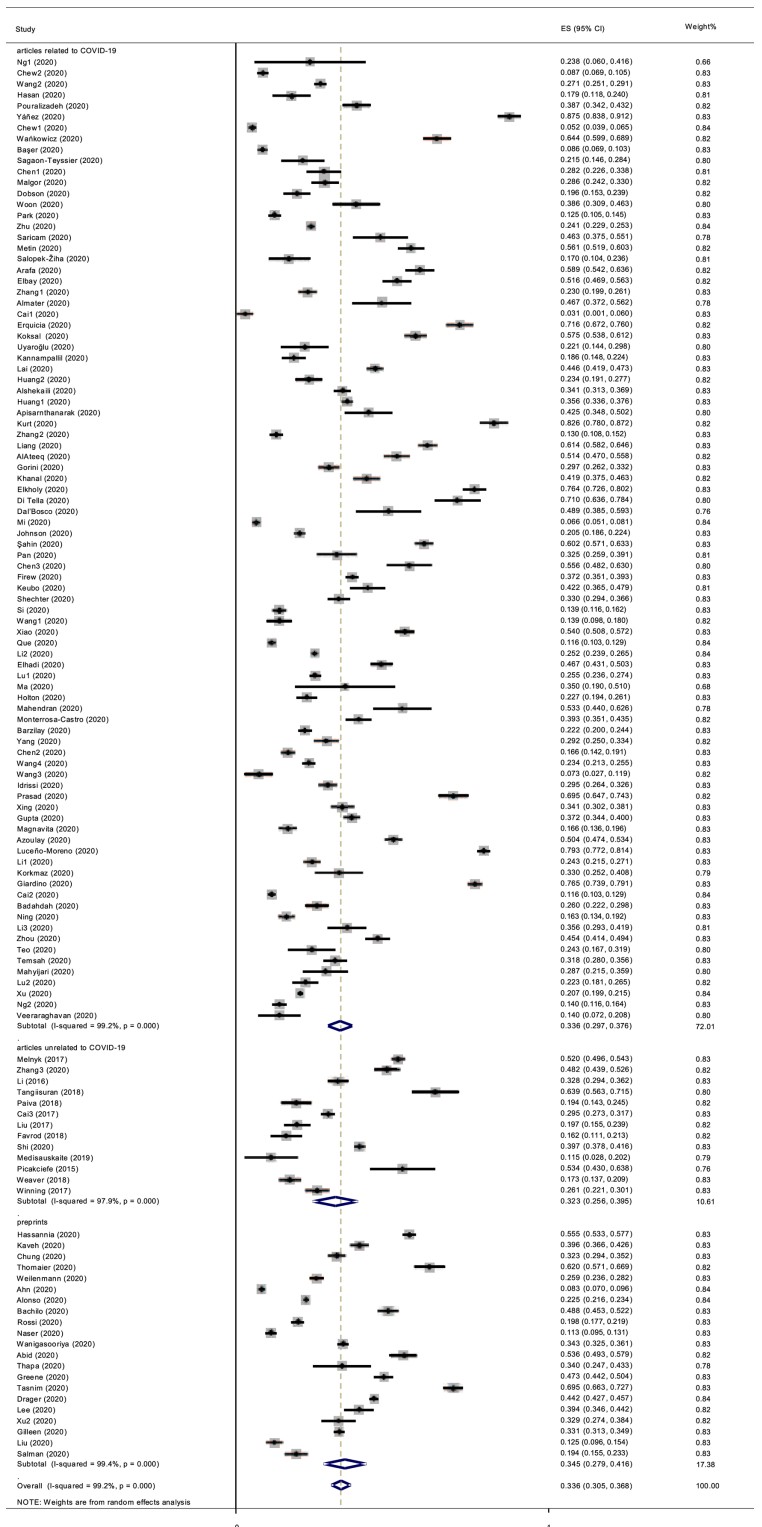

**Figure 2  Forest plot.**

**Table 1 Subgroups analyses.** Univariable meta-regression of included studies: Anxiety.

| Covariate | Number of study | Regression equation β coefficient (95% CI) | P value for heterogeneity |
|---|---|---|---|
| The proportion of female | 118 | −0.1825667 [−0.722132 to 0.356998] | 0.504 |
| The COVID-19-government response stringency index during the survey | 119 | −0.0012402 [−0.005327 to 0.002847] | 0.549 |
| The preparedness of countries in terms of hospital beds per 10,000 people | 119 | 0.003326 [−0.000148 to 0.006800] | 0.060 |
| Human development index | 119 | 0.3003358 [−0.475610 to 1.076282] | 0.445 |

**Table 2 Meta regression.** Subgroup comparison.

| Study type | Number of study (participant) | Combined prevalence | Q | P value for heterogeneity | $I^2$ (95% CI) |
|---|---|---|---|---|---|
| Articles related to COVID-19 | 88 (75,066) | 33.6% [29.7–37.6] | 11,555.59 | <0.001 | 99.2% [99.1–99.3] |
| Articles unrelated to COVID-19 | 13 (9,222) | 32.3% [25.6–39.5] | 559.65 | <0.001 | 97.9% [97.2–98.4] |
| Preprints | 21 (33,737) | 34.5% [27.9–41.6] | 3,377.85 | <0.001 | 99.4% [99.3–99.5] |

## Tests of heterogeneity within subgroups

The heterogeneity in the prevalence of anxiety among health care workers was large and statistically significant ($Q$ = 15,745.82, $p < 0.001$, $I^2$ = 99.2%). Significant heterogeneity was also found in each subgroup (articles related to COVID-19: $Q$ = 11,555.59, $p < 0.001$, $I^2$ = 99.2%; articles unrelated to COVID-19: $Q$ = 559.65, $p < 0.001$, $I^2$ = 97.9%; preprints: $Q$ = 3,377.85, $p < 0.001$, $I^2$ = 99.4%; Table 2). This indicates that whether the study has been published and whether it relates to COVID-19, may not be sources of heterogeneity.

## Sensitivity analysis and publication bias

When each study was serially excluded from the meta-analysis, the combined prevalence did not change significantly. This suggests that the results are relatively stable and reliable (Fig. 3). A funnel plot was drawn to assess the publication bias of the included studies (articles related to COVID-19, articles unrelated to COVID-19, and preprints; Figs. 4–6), the shape of which revealed evidence of asymmetry in articles related to COVID-19 and preprints. This suggests the possibility of publication bias. In addition, a quantitative Egger test was conducted, and the $p$ value for the Egger test was <0.001 in articles related to COVID-19 and <0.05 in preprints. Egger's test revealed no evidence of publication bias in articles unrelated to COVID-19 ($p$ = 0.381). Thus, publication bias exists in articles related to COVID-19 and in preprints.

## DISCUSSION

This meta-analysis aimed to provide quantitative evidence on whether the COVID-19 pandemic increased the prevalence of anxiety among health care workers and whether publication bias existed in these articles during the pandemic. After a comprehensive search, screening, and selection of sources from 2015 to 2020 drawn from six electronic databases, 122 studies were determined to be eligible and were included in this meta-analysis. The effect size was significantly higher than zero in all studies; however,

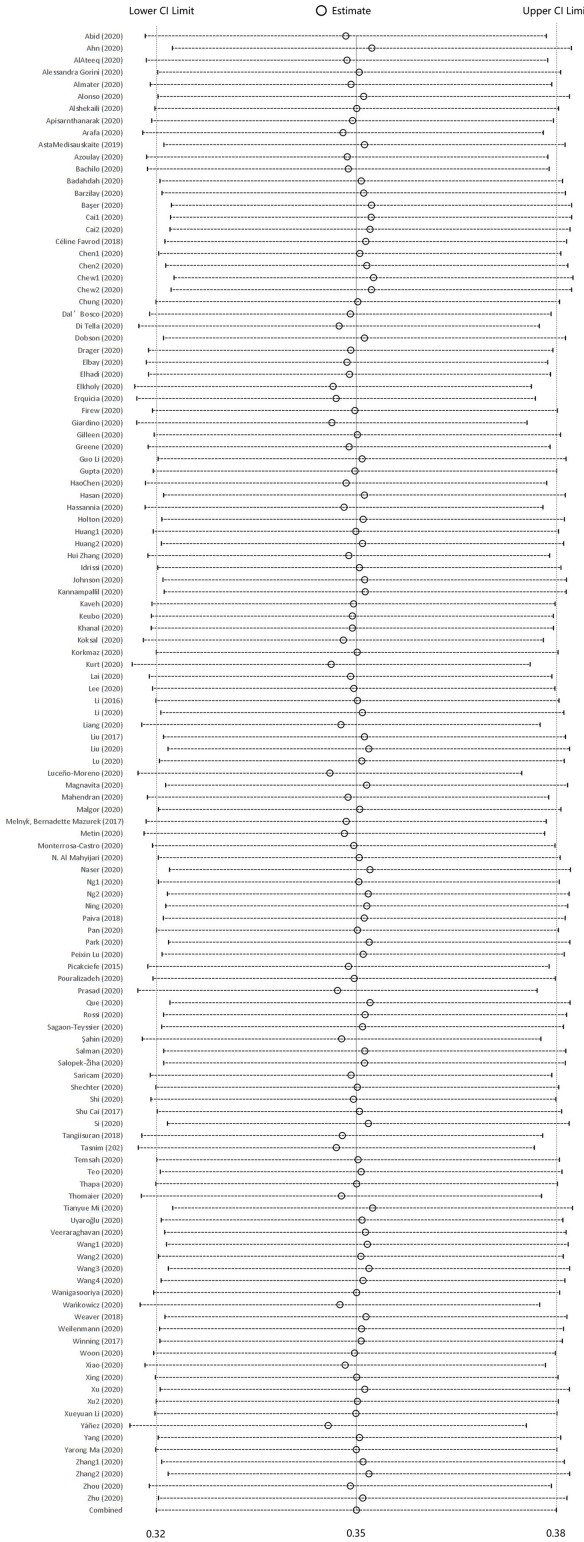

**Figure 3  Sensitivity analysis.**

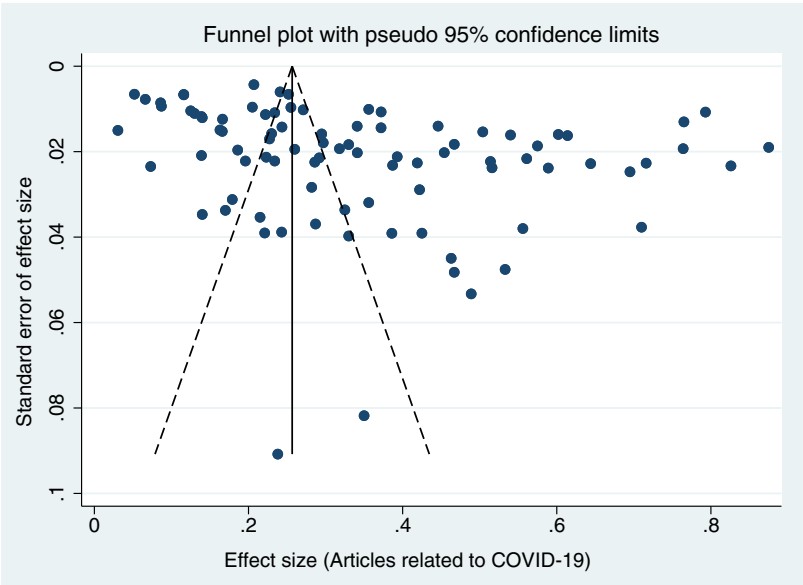

**Figure 4 Funnel plots for publication bias of articles related to COVID-19.**

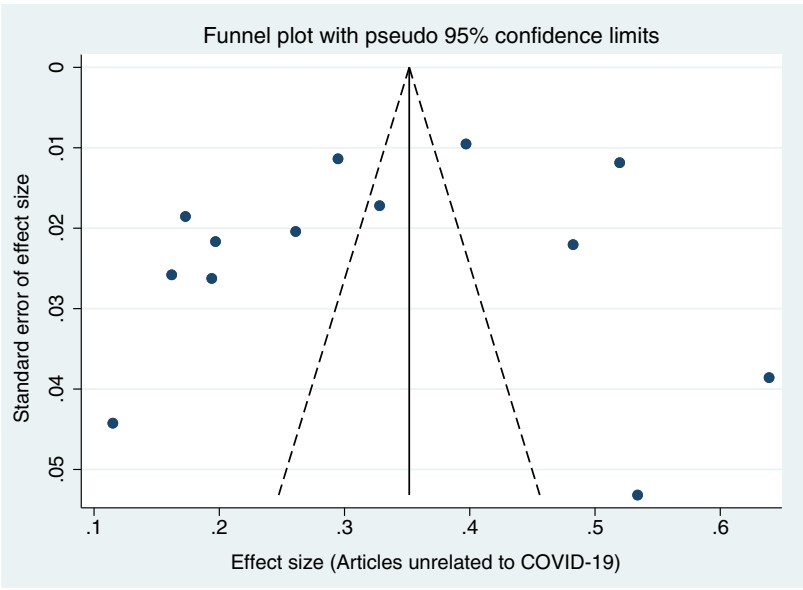

**Figure 5 Funnel plots for publication bias of articles unrelated to COVID-19.**

no significant difference in effect size was found between articles related to COVID-19 33.6% (95% CI [29.7–37.6]) and those that were unrelated 32.3% (95% CI [25.6–39.5]). Thus, the first hypothesis was not supported. Although it appears that the majority of health care workers experienced mild anxiety symptoms, the hypothesis that COVID-19 makes health care workers more anxious was not supported by the quantitative evidence derived in this study. Moreover, no significant difference was found between articles

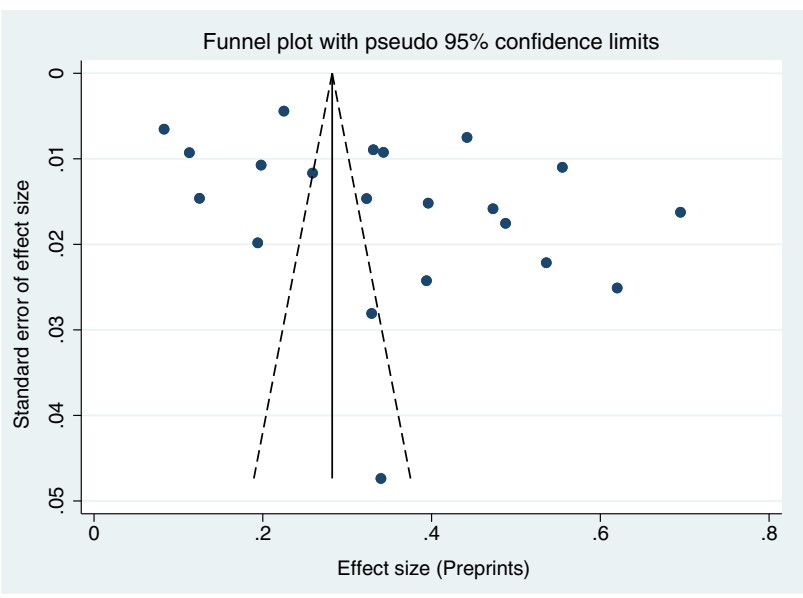

**Figure 6 Funnel plots for publication bias of preprints.**

related to COVID-19 and preprints related to COVID-19. Egger's tests revealed publication bias in both the articles related to COVID-19 and the preprints, supporting the second hypothesis.

The lack of a significant difference in anxiety reported in the pre- and post-COVID-19 literature in the current study may be related to the following factors: health care workers face working overtime at the end of shifts, fear of making mistakes at work, taking unfinished work home, heavy workloads, extended working hours, time conflicts between work and home, conflicts at work with both patients and other staff, and workplace violence (*Bennett et al., 2001*; *Canabaz et al., 2008*; *Fiabane et al., 2013*). All of these factors contribute to much higher risk in comparison to other occupations and may lead to these individuals experiencing a higher level of anxiety than the general population (*Tsai & Liu, 2012*; *Lisspers, Nygren & Söderman, 1997*; *Kader Maideen et al., 2015*; *Henry & Crawford, 2005*). The impact of these factors is likely to be comparable to or even greater than that of COVID-19. Appropriate preparedness of the hospital and the recovery capacity of health care workers may also reduce the impact of COVID-19. For instance, in response to COVID-19, it has been shown that hospitals prepared for an influx of critically ill patients by taking measures such as increasing bed capacity, implementing interdisciplinary cooperation, and preparing adequate medical supplies such as protective clothing and disinfectants to reduce the risk of infection (*Fang, Nie & Penny, 2020*; *Griffin et al., 2020*). In terms of response strategies, Japan, the UK, and the US have invested more health-related resources in the treatment of patients with severe disease, while South Korea and Iran have adopted a strategy of extensive testing and identification of suspected patients (*Wang et al., 2020*). One study showed that apart from a brief initial panic, 99% of health care workers believed that currently available protective measures are sufficient (*Li et al., 2020*). Moreover, a survey of clinical nurses in Saudi Arabia showed

that they have low psychological burden and high resilience (*Balay-Odao et al., 2021*). Therefore, better hospital prevention and good adaptability of health care workers may have led workers to underestimate the impact of COVID-19 and report it as minor. Moreover, some COVID-19-related studies have suggested that the most stressed health care workers may not have time to answer study questionnaires (*Alonso et al., 2020*), which may also lead to an underestimation of anxiety levels during the COVID-19 period. Nevertheless, the present study suggests that more attention should be paid to the mental health of medical staff, and psychological support should be provided, such as remotely delivered psychological therapies, psycho-education, and chatlines.

Our study did not find significant differences in the prevalence of anxiety between COVID-19-related articles and preprints. However, we observed publication bias in both articles related to COVID-19 and preprints (related to COVID-19), which indicates that authors are more likely to submit significant results, whether it is an article or a preprint. Although preprint proponents argue that by mitigating incentives to pursue only significant results, visibility for "negative" studies and unrejected null hypotheses will improve, these claims have yet to be empirically supported (*Nabavi Nouri et al., 2021*). Preprints may not prevent investigators from declining to submit results owing to assumptions on their part that they must have made a mistake, failure to support a known finding, loss of interest in the topic, or anticipation that others will be uninterested in null results. Researchers may only release their studies as preprints if they are confident that the results are appealing. Nevertheless, our findings suggest that caution should be exercised when using preprints as evidence.

The present study deviated from the Stage 1 protocol for several reasons. First, when we collected the data, we found that some articles studying anxiety were situation- and issue-specific, such as virus and death anxiety. Although we declared in the protocol that we would include articles related to anxiety among health care workers, our research aimed to study the general state of anxiety. Therefore, we excluded studies on situation- and issue-specific anxiety. Second, we declared that "if the original paper does not specify the effect size or the number of health care workers with anxiety, the authors of the paper will be contacted and asked to provide this information. If they are unable to do so, the study will be excluded from the analyses." However, we did not set a deadline for this. Ultimately, we excluded studies we could not obtain data for, from the authors after a period of 2 weeks had passed since our contact attempt. Third, we found that we did not consider some key electronic databases, such as Embase and PsycInfo, prior to registration. Thus, we added searches of these two databases based on the Stage 1 protocol. Fourth, because there were too many disparities among the studies (ranging from 3.4% to 87.3%), we performed a Freeman–Tukey double arcsine transformation to stabilize the variance of each study's proportion. Fifth, heterogeneity may have influenced the results. We emphasize that the diagnostic cut-off criteria used were not uniform across the measurement tools in this review; prevalence obtained from the same group using different scales may differ. There may also be differences in the anxiety level of health care workers in countries with different levels of health care; for instance, the anxiety level of health care workers in African countries with relatively low medical standards and
extremely scarce medical supplies may be significantly higher than that of workers in other regions. Previous studies have also revealed that gender, educational background, and age can affect anxiety (*Wang et al., 2017*; *Buyukkececi, 2020*; *Özdin & Bayrak Özdin, 2020*; *Pieh, Budimir & Probst, 2020*). Therefore, we extracted more information and performed additional subgroup analyses and meta-analyses to determine the cause.

This systematic review had several limitations. First, the methods used in the current study may have affected the results. We adopted a random effects model for this analysis. This resulted in articles being greatly influenced by the small sample size, wherein the publication bias was significant, which may have affected our final results. We also used the STROBE checklist to assess the quality of the observational studies in our dataset, similar to a previous study (*Salari et al., 2020*); however, originally, the STROBE was merely a reporting guideline to ensure that the presentation of what was planned and found in an observational study was clear, and it is inappropriate to use the STROBE as an assessment tool to judge study quality (*da Costa et al., 2011*). Therefore, it is necessary to introduce a more appropriate quality assessment base (*e.g.*, the Newcastle–Ottawa Scale) in the future (*Wells et al., 2000*). Second, although we performed a set of subgroup analyses and meta-regression, we failed to explain the heterogeneity in our sample. Third, our results cannot represent the anxiety-related situation during the entirety of the COVID-19 period because we selected studies prior to November 9, 2020, but the COVID-19 pandemic is still ongoing. Accordingly, we have examined is only the results of the studies conducted before November 9, 2020. Fourth, although English is undoubtedly the lingua franca of science, many journals also publish literature in the local language. For example, a non-negligible amount of literature on biodiversity is published in languages other than English (*Nuñez & Amano, 2021*). The present study included only studies written in English; however, this English-restricted approach is certainly a limitation in terms of clarifying the reality of stress among health care workers.

The unequal distribution of studies in different regions, limited collection time, and language limits may have affected the external validity of our study. In the future, we hope that more research can be conducted in countries with poorer medical conditions to make the scope of research more comprehensive and that more studies can be included to determine the factors affecting anxiety among health care workers and improve external validity.

## CONCLUSIONS

Our study found no significant differences in effect sizes (prevalence of anxiety) among studies related to COVID-19 and those that were unrelated. Whether the state of health care workers' anxiety is altered by the COVID-19 pandemic is currently difficult to assess. However, there is evidence that their anxiety levels may always be high, which suggests that more attention should be paid to their mental health. Furthermore, we found a large publication bias among the studies in our analysis; however, the quality of the studies appears to be relatively stable and reliable.

# ACKNOWLEDGEMENTS

We would like to thank Editage for editing and reviewing this manuscript for English language.

### Funding

This research was supported by the Japan Society for the Promotion of Science (https://www.jsps.go.jp/english/) KAKENHI Grant Numbers JP16H03079 (Yuki Yamada), JP17H00875 (Yuki Yamada), JP18K12015 (Yuki Yamada), JP20H04581 (Yuki Yamada), and JP21H03784 (Yuki Yamada). The funders had no role in study design, data collection and analysis, decision to publish, or preparation of the manuscript.

### Grant Disclosures

The following grant information was disclosed by the authors:
Japan Society for the Promotion of Science KAKENHI: JP16H03079, JP17H00875, JP18K12015, JP20H04581 and JP21H03784.

### Competing Interests

Yuki Yamada is an Academic Editor for PeerJ.

### Author Contributions

- Lunbo Zhang conceived and designed the experiments, performed the experiments, analyzed the data, prepared figures and/or tables, authored or reviewed drafts of the paper, and approved the final draft.
- Ming Yan conceived and designed the experiments, performed the experiments, prepared figures and/or tables, authored or reviewed drafts of the paper, and approved the final draft.
- Kaito Takashima conceived and designed the experiments, authored or reviewed drafts of the paper, contacted the included studies' authors, and approved the final draft.
- Wenru Guo conceived and designed the experiments, authored or reviewed drafts of the paper, contacted the included studies' authors, and approved the final draft.
- Yuki Yamada conceived and designed the experiments, authored or reviewed drafts of the paper, provided funding acquisition, and approved the final draft.

### Data Availability

The raw data is available in the Supplemental File.

### Supplemental Information

Supplemental information for this article can be found online at http://dx.doi.org/10.7717/peerj.13225#supplemental-information.

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
