# Peer review of "The effect of the COVID-19 pandemic on health care workers’ anxiety levels: a meta-analysis"

_PeerJ, doi:10.7717/peerj.13225_

## Round 0.1 · original submission · Major Revisions

The decision is to Revise.

Reviewer 1 ·

Basic reporting

The following major issues were identified in this manuscript
- Method of selection is not well documented

Experimental design

- Statistical method is without necessary details to allow replicability of the work, you need to explain the statistical method in further details. You need to use other statistical tests to support the study findings or to disprove them.

Validity of the findings

- Heterogeneity might be a factor in the result and should be acknowledged

Additional comments

- Introduction and discussion are not well constructed and need to cite recent literature.

·

Basic reporting

#1.1 Generally, the authors should follow and report in line with an updated PRISMA 2020 version.
- Page MJ, et al. The PRISMA 2020 statement: an updated guideline for reporting systematic reviews. BMJ. 2021 Mar 29;372:n71.

#1.2 Please provide a Table that summarizes the characteristics of the included studies such as country or geographical region based on WHO/World Bank income groups, study design, data collection, measurement tools, diagnostic cut-off that used for anxiety assessment, number of participants, mean age, the proportion of female, marriage status, working position (frontline vs. second-line), type of hospital (primary, secondary, tertiary), professional role (physicians, nurse, others).

#1.3 To shed light on the impact findings with regard to the prevalence of mental health problems during the COVID-19 pandemic, under the “Discussion” section, I advocate the authors should be discussed in terms of the hospital preparedness and resilience capability (e.g., hospital safety, hospital emerging disease leadership and cooperation, hospital emerging disease plan, emergency stockpiles and logistics management, and emergency staff and capability) to confront during the situation of disasters or public health infectious events.

Moreover, comprehensive discussion and compare the findings to other populations that facing the COVID-19 pandemic would be interest for readers as well as the scientific community as follows:
- What We Have Learned from Two Decades of Epidemics and Pandemics: A Systematic Review and Meta-Analysis of the Psychological Burden of Frontline Healthcare Workers. Psychother Psychosom. 2021;90(3):178-190.
- Global prevalence of mental health issues among the general population during the coronavirus disease-2019 pandemic: a systematic review and meta-analysis. Sci Rep. 2021 May 13;11(1):10173.
- Prevalence of mental health problems among children and adolescents during the COVID-19 pandemic: A systematic review and meta-analysis. J Affect Disord. 2021 Oct 1;293:78-89.

#1.4 For statistical reporting, the I2 index should be expressed along with the 95% CI through the main text and figures.

#1.5 Table 1 Subgroup analyses, should be reported along with the P for interaction and the number of participants included.

Experimental design

#2.1 The search strategies should be updated to identify all available evidence in the recent year, they need to be included in this analysis. Also, key electronic databases including Embase, PsycInfo, and gray literature should be included.

Validity of the findings

#3.1. Under “Statistical Analysis”, the statistical approach for pooled effect estimates was not clear. Specifically, the random-effects model was performed based on a normal distribution or account for the effects of studies with extreme (small or large prevalence estimates), in which the normal approximation procedures often break down.

Theoretically, based on the broad definition of the methodological approach across the included studies as well as the disparities between countries that may affect the prevalence of anxiety, I recommended the authors perform a meta-analysis with appropriate methods which dealing with proportions close to or at the margins before pooling the effect estimates. Using the binomial distribution to model the within-study variability or by allowing Freeman-Turkey double arcsine transformation to stabilize the variances has been suggested.

#3.2. To shed light on the impact findings, under the “Statistical Analysis” section, both methodological and statistical heterogeneity should be comprehensively explored. I advocate the authors to performed subgroup analysis and meta-regression based on country or WHO regions with respect to the key characteristics of the included studies or major factors that may affect the preparedness to COVID-19 infection or associated with the classification of being anxiety (e.g., the proportion of female, working position [frontline vs. second-line], type of hospital [primary, secondary, or tertiary], professional role [physicians, nurse, or others], study design [cross-sectional, cohort, longitudinal survey], data collection [online survey, telephone, or paper-based survey], measurement tool [GAD-7, BDI-Anxiety, HADS, or DASS-21, etc.] and diagnostic cut-off).

Moreover, subgroup analysis and meta-regression according to the disparities between countries in terms of the poverty impacts of COVID‑19, preparedness of countries to respond, and economic vulnerabilities that impact the prevalence of anxiety should also be performed (e.g., the COVID-19-government response stringency index during the survey, the preparedness of countries in terms of hospital beds per 10,000 people, World Bank Countries by income groups, Human development groups, etc.).

---

## Round 0.2 · Major Revisions

The decision is to Revise.

·

Basic reporting

The English language used is clear. The structure of the article complies with the PRISMA guidelines. Raw data has been provided.

Comment 1 - The introductory section could be implemented by explaining more clearly which topic was covered in the literature considered for the subgroup "unrelated COVID-19". It would be appropriate to indicate why the authors chose this topic for comparison and not literature from previous outbreaks, which might represent stress situations similar to that due to COVID but on a smaller scale. perhaps the authors' intention was to compare anxiety levels with different situations, to highlight the actual pandemic from COVID-19?

Comment 2 - The figures are relevant and of high quality, but I suggest improving the labels and changing the PRISMA checklist to the updated version (2020) found at http://prisma-statement.org/PRISMAStatement/Checklist.

Experimental design

The research question was relevant and meaningful. The research was well executed, in line with PRISMA guidelines.

Comment 3 - The methods were described with sufficient detail and information to replicate the study. However, I urge the authors to clarify what is stated in the commentary on the introductory section, as only by reading the exclusion criteria (specifically 4 - publications about other outbreaks) does the focus of the unrelated COVID-19 articles become clear.

Validity of the findings

The results are clearly reported, and the discussion appears comprehensive.

Comment 4 - I would suggest that the authors also comment briefly in this section on the choice not to compare the results of the COVID-19 studies with a subset of studies on other epidemics, and to specify what the studies in the unrelated COVID-19 subgroup were about.

Additional comments

Thank you for the opportunity to review this work. The study is interesting and deals with an important issue. The paper is generally clear and complete. I understand the long work done, in the study and in the paper, however some minor revisions and explanations of the authors are necessary (please see Comment 1, 2, 3, 4).

Reviewer 4 ·

Basic reporting

This is an important study in an under-researched area of the world. The literature addressed is described accurately so far as I can see. The method seems to have been followed faithfully and the authors were well-positioned to conduct the analysis. The material is interesting, and the topic is relevant. However, some areas need clarifications:
- Describe more deeply the rationale for the review in the context of what is already known. What is the study's biggest contribution? The contribution should be clearly stated in the background.

Experimental design

- Specify the date when each source was last searched or consulted.
- State the process that will be used for selecting studies through each phase of the review (that is, screening, eligibility and inclusion in meta-analysis).
- How do you deal with missing outcome data? The researchers will contact and request first authors through electronic mails to provide missing outcome data, perform sensitivity analysis to assess the robustness of meta-analytic results, and discuss the potential impact of missing data on the review findings. Do you use any one of the several statistical approaches for dealing with missing outcome data?
- Please provide more information about how you perform sensitivity analysis to reflect the extent to which the meta-analytical results and conclusions are altered as a result of changes in the analysis approach.
- After performing meta-analysis, the researchers should be computing prediction interval to reflect the variation of outcomes of anxiety in different settings, including the direction of evidence in future studies.
- Please report the software package(s) used.

Validity of the findings

- The discussion section should be reorganized because they are poor. I believe there should be better integration of the results with the existing literature. Discuss the generalisability (external validity) of the study results. Also, discuss both the direction and magnitude of any potential bias.
- The recommendations for practice/research/education/management should have been approached in greater depth.

Additional comments

The manuscript will serve a broad audience of students, researchers, and practitioners, however, the manuscript needs to be carefully and attentively proofread because some sentences are awkwardly constructed, punctuation is deficient, and therefore reading is occasionally difficult to follow. The English of this manuscript should be reviewed by a fluent English speaker.

---

## Round 0.3 · accepted · Accept

The authors have improved the quality of the manuscript by taking into account the referees' comments.

·

Basic reporting

No comment

Experimental design

No comment

Validity of the findings

No comment

Additional comments

Thank you for the opportunity to review this work again. The paper is generally clear and complete the authors have improved the quality of the manuscript, taking into account my comments.

Reviewer 4 ·

Basic reporting

The material is interesting and the topic is relevant.

Experimental design

Methods well described.

Validity of the findings

The conclusions are well stated and linked with the research question.

Additional comments

I believe that the review carried out has greatly improved the quality of the study. Also, I do think that the author(s) addresses the broad questions, appropriately which were asked. Congratulations!

External reviews were received for this submission. These reviews were used by the Editor when they made their decision, and can be downloaded below.